# RoboPhD: Self-Improving Text-to-SQL Through Autonomous Agent Evolution

## Abstract

We present RoboPhD, a system where AI agents autonomously conduct research to improve Text-to-SQL performance. RoboPhD implements a closed-loop evolution cycle with two coordinated components: a SQL Generation agent composed of a database analysis script and SQL generation instructions, and an Evolution agent that designs new versions based on performance feedback. Central to the framework is an ELO-based selection mechanism enabling survival-of-the-fittest dynamics while handling non-transitivity in performance.

Starting from a naive 70-line baseline, RoboPhD evolves agents through iterative cross-pollination, discovering effective techniques without any external guidance on the Text-to-SQL domain. Our best agent, evolved to 1500 lines over 18 iterations, autonomously discovered strategies such as size-adaptive database analysis that adjusts depth based on schema complexity and SQL generation patterns for column selection, evidence interpretation, and aggregation.

Evolution provides the largest gains on cheaper models: while we improve by 2.3 points over a strong Claude Opus 4.5 naive baseline, we show an improvement of 8.9 points over the weaker Claude Haiku model. This enables "skip a tier" deployment: evolved Haiku exceeds naive Sonnet accuracy, and evolved Sonnet exceeds naive Opus—both at lower cost.

The full system achieves 71.3% accuracy on the BIRD development set, demonstrating that AI can autonomously build a strong agentic system without human intervention.

## 1 Introduction

Text-to-SQL, the task of translating natural language questions into executable SQL queries, remains a fundamental challenge in natural language processing with significant real-world applications. Recent advances have shown that large language models (LLMs) can achieve impressive performance on benchmarks like Spider (Yu et al., 2018) and BIRD (Li et al., 2023), but these approaches typically require extensive manual prompt engineering, careful model selection, and domain expertise.

The ability of AI systems to conduct AI research autonomously has been identified as a crucial capability milestone with profound implications for the pace of AI progress (Steinhardt, 2022; Cotra, 2022). Several researchers have argued that AI systems capable of improving themselves or conducting research represent a potential discontinuity in technological development (Bostrom, 2014; Yudkowsky, 2008). Recent work has begun to explore this frontier, with systems demonstrating increasing autonomy in scientific discovery (Romera-Paredes et al., 2024), machine learning research (Lu et al., 2024), and algorithm development (Fawzi et al., 2022).

While well short of a full self-improving system, our work demonstrates a concrete instantiation of AI agents autonomously conducting the research cycle of hypothesis generation, experimentation, and iterative refinement to improve performance on text-to-SQL generation.

RoboPhD's architecture evolves AI agents composed of two artifacts—a database analysis tool and SQL generation instructions—using an architecture that separates offline analysis from online inference. This pattern generalizes to other tasks with similar structure.

This work makes the following key contributions:

- **Autonomous AI Research Framework**: We demonstrate AI agents conducting systematic research on text-to-SQL through a closed-loop evolution cycle, autonomously discovering effective optimization strategies without human intervention

- **ELO-Based Evolutionary Selection**: First application of ELO ratings for evolutionary prompt/agent optimization, effectively handling non-transitivity and asynchronous agent entry

- **Evolution of a complete Text-to-SQL system**: The framework simultaneously evolves a set of Python database analysis tools and SQL-generation instructions to improve end-to-end performance.

- **Inverse Scaling of Evolutionary Benefit**: We show that evolution provides larger gains on cheaper models (+8.9 points for Haiku vs +2.3 for Opus), enabling cost-effective deployment where evolved cheaper models exceed naive expensive models

- **Self-Directed Improvement without Author-Supplied Domain Expertise**: The system improves without author-provided Text-to-SQL knowledge from a trivial baseline; gains arise from latent LLM capabilities and iterative error-driven learning across iterations

## 2 RELATED WORK

### 2.1 TEXT-TO-SQL SYSTEMS

The BIRD benchmark (Li et al., 2023) has driven significant advances in Text-to-SQL, with top systems achieving (as of mid-September 2025) 71-77% accuracy through diverse approaches. AskData (Shkapenyuk et al., 2025) achieves 77.14% using GPT-4o with data analysis agents. CHASE-SQL (Pourreza & Rafiei, 2024) reaches 76.02% through multi-path reasoning and preference optimization. XiYan-SQL (Liu et al., 2024) employs multi-scale few-shot learning for 75.63% accuracy. CSC-SQL (Sheng et al., 2025) uses community-driven schema construction achieving 73.67%. Reasoning-SQL (Pourreza et al., 2025) distills capabilities to smaller models reaching 72.78%. OpenSearch-SQL (Xie et al., 2025), OmniSQL (Li et al., 2025a), and GenaSQL (Dönder et al., 2025) all achieve approximately 72% through various architectures. Distillery (Maamari et al., 2024) introduces compositional approaches at 71.83%. CHESS (Talaei et al., 2024) achieves 71.1% using contextual harnessing.

Our work uniquely focuses on *autonomous discovery* of these strategies rather than manual design. While prior systems require human experts to engineer prompts and architectures, RoboPhD discovers optimization techniques autonomously. Note that we have experimented with using all of the above papers (with the exception of the BIRD benchmark paper (Li et al., 2023)) as inputs to the evolution agent in the *research-driven* evolution strategy mentioned in Section 5.2.

Prior work on prompt optimization includes APE (Zhou et al., 2022), OPRO (Yang et al., 2023), DSPy (Khattab et al., 2023), and TextGrad (Yuksekgonul et al., 2025). While these systems optimize prompts, our approach evolves complete agent artifacts including database analysis tools and SQL generation instructions. Crucially, our system conducts research autonomously without human intervention in the optimization loop. One key aspect of our approach is allowing the Evolutionary agent access to the errors made from previous candidate agents. This is similar to the idea of *text gradients* explored in the TextGrad paper (Yuksekgonul et al., 2025), where the LLM produces a (metaphorical) gradient to describe how to address a particular error. In our system, the error gradients are available for the agent to choose how to incorporate them into the next candidate agent.

Evolutionary algorithms have been applied to neural architecture search (Pham et al., 2019) and hyperparameter optimization (Loshchilov & Hutter, 2016; Jaderberg et al., 2017). We extend this to evolving complete AI agents with natural language specifications and tool usage, creating a fully autonomous research system.

The ELO rating system (Elo, 1978), originally developed for chess, has found recent applications in AI evaluation. The Chatbot Arena (Zheng et al., 2023) uses ELO ratings to create a dynamic leaderboard of LLMs based on human preferences, demonstrating ELO's effectiveness for model comparison. While these systems use ELO for passive evaluation and ranking, we introduce ELO as an active selection mechanism for evolutionary optimization (see §3.2.5 for our implementation).

# 3 METHOD

Figure 1 shows the high-level architecture of RoboPhD. The system takes three inputs: a simple *naive agent* package containing minimal Text-to-SQL logic (∼70 lines total), an *evolution strategy* that provides meta-level research guidance, and the BIRD *training database* (69 databases, 6.6K training questions). Through competitive evolution over $N$ iterations, these simple origins yield a production-ready agent. Our best agent grew from 50 lines of code and 20 lines of SQL generation instructions to over 1000 lines of evolved code and 500 lines of generation instructions across 18 evolutionary iterations, achieving 71.3% accuracy on the BIRD development set.

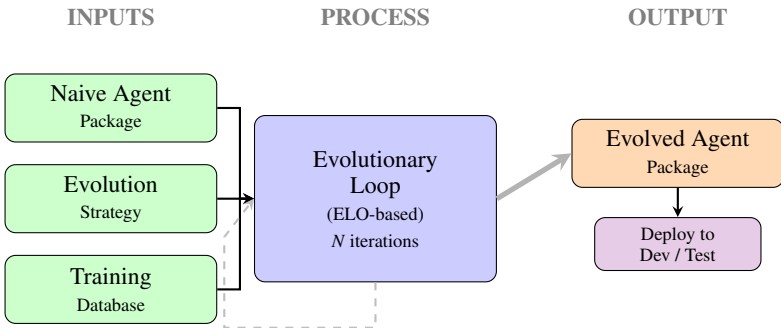

Figure 1: RoboPhD system overview. Simple inputs (a naive agent, an evolution strategy, and training data) feed into an ELO-based evolutionary loop that produces a strong, production-ready agent for deployment.

## 3.1 CONCEPTUAL OVERVIEW: THE GRADUATE STUDENT METAPHOR

RoboPhD operates like a graduate student operating with minimal faculty supervision, focused on reducing error rate. This paper describes an application to the BIRD benchmark, but the lack of SQL-specific initial guidance points towards easy transfer to other domains. The evolutionary agent, here implemented as Claude Code[1], wrapping Claude Sonnet 4.5[2] or Opus 4.5[3], can be interpreted as a doctoral student attempting to build a complete AI-powered text-to-SQL system, analogous to prior efforts on the BIRD benchmark.

The graduate student (AI Evolutionary Agent) analyzes experimental failures to identify patterns, forms hypotheses about improvements, and implements these ideas as new system designs. System designs are tested and refined at least once before being deployed. Each agent it creates consists of a Python-based database analysis tool and SQL generation instructions for the evaluation model. Note that, while we have a new "graduate student" with each iteration, the evolutionary agent maintains context across the entire process of creating an agent package, refining its design and accumulating expertise with each system test.

A core design goal of RoboPhD is *domain independence*: we intentionally avoid injecting author-crafted Text-to-SQL techniques. The evolution strategy (see Appendix C) provides only meta-level research direction ("combine the best techniques from top-performing agents"), rather than pre-scribing scientific content about Text-to-SQL itself. Consequently, measured gains arise from two sources: (i) the latent capabilities of the underlying LLMs and (ii) experience accumulated over many iterations as the Evolutionary Agent builds successively stronger agents that it refines and combines to create even stronger agents.

Algorithm 1 presents the complete evolution cycle, showing how strategy selection, agent evolution, evaluation, and ELO updates work together to drive continuous improvement.

---

[1] https://claude.com/product/claude-code

[2] https://www.anthropic.com/claude-sonnet-4-5-system-card

[3] https://www.anthropic.com/claude-opus-4-5-system-card

---

**Algorithm 1** RoboPhD Evolution Cycle

---

1: **Input:** Database pool $\mathcal{D}$, question pool $\mathcal{Q}$, iterations $N$, strategy $S$, initial agents $\mathcal{A}_0$
2:     // *In this paper:* $\mathcal{D}$ = *BIRD train (69 databases),* $S$ = *cross-pollination,* $\mathcal{A}_0$ = *[naive]*
3: **Output:** Final agent rankings and performance history
4:
5: $\mathcal{A} \leftarrow \mathcal{A}_0; competitors \leftarrow \mathcal{A}_0$
6:
7: **for** $i = 1$ to $N$ **do**
8:     // *Evaluation*
9:     $databases \leftarrow \text{RandomSample}(\mathcal{D}, 5)$
10:    **for** each $db$ in $databases$ **do**
11:        $questions[db] \leftarrow \text{RandomSample}(\mathcal{Q}[db], 30)$
12:    **end for**
13:    **for** each $agent$ in $competitors$ **do**
14:        $accuracy[agent] \leftarrow \text{EvaluateOnQuestions}(agent, databases, questions)$
15:    **end for**
16:
17:    // *ELO Update via Pairwise Decomposition*
18:    **for** each pair $(a, b)$ in $\{(1, 2), (1, 3), (2, 3)\}$ **do**
19:        $score \leftarrow 1$ if $accuracy[a] > accuracy[b]$, 0.5 if equal, else 0
20:        $\text{UpdateELO}(competitors[a], competitors[b], score, K = 32)$
21:    **end for**
22:
23:    $winner \leftarrow \arg\max_{agent \in competitors} accuracy[agent]$
24:
25:    // *Prepare next iteration*
26:    **if** $i < N$ **then**
27:        $agent_{new} \leftarrow \text{EvolveAgent}(competitors, S, \text{ErrorAnalysis}(i))$
28:        $\mathcal{A} \leftarrow \mathcal{A} \cup \{agent_{new}\}$
29:
30:        // *Tournament Selection for Next Round*
31:        $competitors[1] \leftarrow winner$ {Current winner}
32:        $competitors[2] \leftarrow agent_{new}$ {Newly evolved}
33:        $competitors[3] \leftarrow \text{RandomTop}(\mathcal{A}, 2)$ {*Random selection from top 2 by ELO*}
34:    **end if**
35: **end for**
36: **return** $\mathcal{A}$ with final ELO rankings

---

### 3.2 TECHNICAL IMPLEMENTATION

We now detail how RoboPhD achieves accurate SQL generation from simple initial components through an autonomous evolution process.

#### 3.2.1 SYSTEM OVERVIEW: EVOLVING AN AGENT WITH TWO COMPONENTS

RoboPhD's architecture centers on an *agent* consisting of two components that are evolved together each iteration, as illustrated in Figure 2:

1. **Database Analysis Tool**: A deterministic Python script that analyzes database schemas offline, producing structured documentation of tables, relationships, sample values, and data patterns.

2. **Eval Instructions**: SQL generation guidance that teaches the LLM how to transform natural language questions into precise SQL queries.

These artifacts operate in sequence: the analysis tool examines a database offline, producing structured output that feeds into online SQL generation. The SQL generation model never sees the database schema directly—all understanding must flow through the analysis layer, forcing the evolution process to develop effective communication strategies between its components.

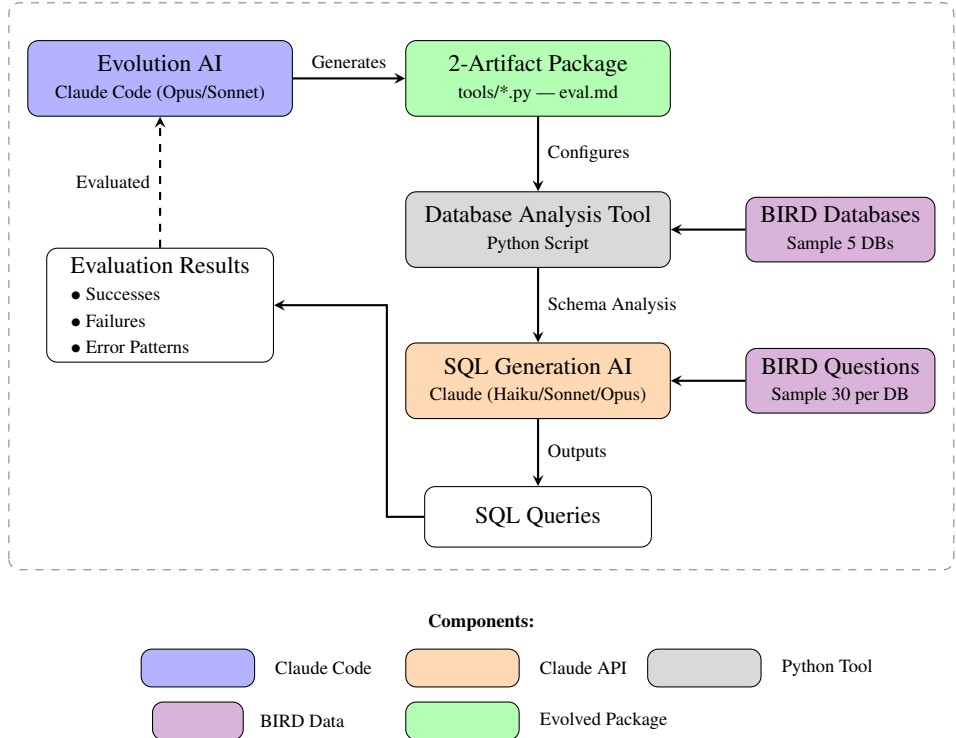

Figure 2: The RoboPhD evolutionary cycle. The Database Analysis Tool (gray) is a deterministic Python script, not an LLM call, making offline analysis fast ($0.00) and reproducible.

### 3.2.2 OFFLINE DATABASE ANALYSIS PHASE

The database analysis phase is conducted without seeing the questions that will be asked. This separation is typical of industrial requirements where extensive database analysis can be performed offline to facilitate SQL generation, which operates under tight time constraints. This pattern generalizes broadly: separating offline analysis from online inference enables thorough preprocessing to support rapid real-time responses.

**Tool-Only Execution:** Our best-performing approach uses deterministic Python scripts that analyze databases without any LLM calls. The evolved tool receives full access to a SQLite database (schema and row data) and produces structured analysis including (for our better performing agents): complete DDL schema, table relationships with foreign key cardinality, sample values for each column, enumerated values for categorical columns, and data format patterns (dates, currencies, codes).

We experimented with three approaches during system development: (1) LLM-only analysis where an AI agent reasons about the schema, (2) hybrid approaches where an LLM orchestrates Python tools and synthesizes their outputs, and (3) tool-only execution where Python scripts generate complete analysis deterministically. Through competitive evaluation, tool-only execution emerged as superior for two reasons: it is fast and economical ($0.00 per database vs. $0.50 for LLM-based analysis), and, perhaps because a deterministic target is easier to optimize in an evolutionary cycle, it yielded our strongest results. The tool-assistend hybrid approach remains as a fallback mechanism if the evolved Python tool fails, but in practice this is relatively rare. For instance, tool failures accounted for only 5.5% of total costs in the evolution of our best-performing system.

### 3.2.3 ONLINE SQL GENERATION PHASE

The goal of a Text-to-SQL system is to transform natural language questions into executable SQL queries. The SQL generation phase is the end of the process, where the system transforms natural language questions into executable SQL through concatenation of the two evolved artifacts with user inputs:

$$\text{Prompt} = \text{DatabaseAnalysis} \oplus \text{EvalInstructions} \oplus \text{Question} \oplus \text{Evidence}$$

where $\oplus$ denotes concatenation. DatabaseAnalysis is the output of the evolved Python tool (Section 3.2.2). EvalInstructions is the evolved SQL generation guidance. The Question is the natural language question as provided by the user (here, BIRD benchmark). Evidence is provided by BIRD and supplies hints about the domain/schema and clarifications for the question.

**The Role of Eval Instructions:** The `eval_instructions.md` file serves as the bridge between database analysis and SQL generation. While the analysis tool extracts database structure, the eval instructions guide how to transform natural language questions into precise SQL queries. These instructions operate at the final online inference step. Successful eval instructions balance comprehensiveness with precision. Overly lengthy instructions can be counterproductive, as the LLM may have difficulty focusing on the most important components and they may use up too much of Claude's 200K token limit.

Our best-performing agent's eval instructions (527 lines) include: (1) **Column Selection Discipline**: Detailed rules with examples showing that, for instance, "Which movie has the best rating?" should return only the title, not `SELECT Title, Rating`—a critical requirement where extra columns count as failure. (2) **Evidence Interpretation Patterns**: Seven distinct patterns for parsing BIRD's evidence field, from column mappings ("full address refers to street_num, street_name, city") to complex percentage formulas. (3) **String Matching Rules**: Instructions to check the tool's enum value output for exact case-sensitive matches, with guidance on when to use `LIKE` vs. exact equality. (4) **Percentage Formula Parsing**: Rules distinguishing "percentage of A that are B" from "percentage of B that are A"—the word order determines which count is the numerator vs. denominator.

**Agentic Query Result Validation**: text-to-SQL systems face two critical failure modes: syntax errors and semantic errors. Syntax errors prevent execution, while semantic errors produce incorrect results. Prior work typically addresses these challenges through multiple generation attempts. For instance, OpenSearch-SQL (Xie et al., 2025) generates multiple candidate queries and uses self-consistency voting. Our system employs an agentic answer evaluation approach:

**Universal Verification**: Every generated SQL query undergoes verification where the model reviews its own query execution results in the context of the question and the system prompt with database analysis information. The model then responds with either `CORRECT` to accept those results as the answer or an improved SQL query in light of this new information from the candidate execution result. This self-verification and correction process can iterate up to $k = 2$ times, using progressive temperature ($0.0 \rightarrow 0.2 \rightarrow 0.3$) to encourage exploration on retries. Additionally, if iteration $k$ produces errors or an empty (null) result, the system employs an additional targeted retry with an alert highlighting the SQL error or null result.

### 3.2.4 EVOLUTION: CREATING NEW ARTIFACTS

The Evolution AI creates new artifacts based on comprehensive performance feedback from previous iterations (see Algorithm 1). It receives ELO rankings, accuracy metrics, and detailed error analysis, then generates two artifacts that define a candidate solution: 1) a database analysis tool (`tools/*.py`); and 2) SQL generation instructions (`eval_instructions.md`). The evolution process is guided by a hand-crafted strategy prompt; evolving these strategies themselves is future work (Section 5.2).

**Cross-Pollination Strategy:** Our primary evolution strategy instructs the Evolution AI to examine top-performing agents and combine their best techniques into a new "super-tool." The strategy prompt (see Appendix C) emphasizes:

- **Identifying complementary strengths**: Which techniques do different agents use effectively? How do they complement each other?

- **Error-driven improvement**: What error patterns appear in the iteration's `error_analysis_report.md`? How can the new agent address them?

- **Tool-only requirement**: The strategy requires deterministic Python tools rather than LLM-based analysis, ensuring speed and reproducibility.

The Evolution AI produces a `reasoning.md` file documenting its analysis before generating artifacts, providing transparency into its design decisions. We experimented with alternative strategies (refinement, research-driven, error-focused) but found cross-pollination with tool-only emphasis consistently produced our strongest agents.

**Deep Focus Refinement:** Before a newly-evolved agent is put into the main competitive loop, it undergoes a multi-round testing process we call "Deep Focus." After generating an artifact using the methodology described above, the system tests the newly created agent on the questions and databases from prior iterations, beginning with the most recent iteration and working backwards, comparing it with the corresponding agents from those iterations (for instance, it is shown questions the new agent uniquely answered correctly and uniquely answered incorrectly). After each testing round, the Evolution AI can refine its artifacts based on this performance feedback. We parameterize this with $k$ test rounds (default $k = 1$), where each round tests against one additional prior iteration. Our experiments with $k \in \{0, 1, 2, 3\}$ found that even a single test round ($k = 1$) provides meaningful refinement opportunities while keeping evolution time reasonable. Key to Deep Focus is that all Deep Focus testing and refinement takes place in the same Claude Code session in which the agent was created, enabling the Evolution AI to maintain continuity of planning and analysis.

### 3.2.5 Evaluation and ELO Ranking

The system evaluates three agents simultaneously on identical sets of databases and questions, computing overall accuracy using BIRD's set-based comparison (row order ignored, exact match required). Each iteration tests agents on randomly selected databases with their associated questions, providing head-to-head comparisons on the exact same tasks.

These three-way competitions are decomposed into three pairwise comparisons. For example, if agents achieve 65%, 62%, and 62% accuracy, we process three head-to-head results: A beats B (1-0), A beats C (1-0), and B ties C (0.5-0.5).

Maintaining an accurate picture of relative agent strength is critical for the evolutionary algorithm. This work introduces ELO as a ranking mechanism for evolutionary algorithms, updating scores using the standard formula ($\Delta_{ELO} = K(S - E)$, $K = 32$). ELO offers key advantages for our setting:

(1) **Asynchronous Competition**: Like chess players who play their first tournament match at different times, agents can join the population at any point while maintaining fair comparisons through persistent ratings. ELO naturally weights unexpected outcomes more strongly than expected outcomes—when a low-rated (or newly-created) agent defeats a high-rated one, both scores change much more than if the higher-rated agent defeats the lower-rated.

(2) **Non-transitivity Handling**: ELO naturally accommodates rock-paper-scissors dynamics where on one sample of databases and questions, Agent A beats B, on another sample B beats C, but on a third sample C beats A—a situation which is common when different agents have different strengths and weaknesses.

(3) **Task Normalization**: Win/loss treatment normalizes across varying difficulty—although accuracy commonly swings from 60% to 80% on different database samples, relative rankings on the same task remain stable.

This competitive framework enables survival-of-the-fittest dynamics while maintaining fairness across agents with different entry points.

### 3.2.6 Experimental Protocol: Selection and Sampling

**Agent Selection:** In a standard evolutionary iteration, (see Appendix D for details on exceptions), RoboPhD tests three agents selected via a priority system: (1) the previous winner, (2) the newly evolved agent, and (3) a random agent selected from the top 2 performers by ELO (other than the winner). This balance ensures thorough testing while maintaining the evolutionary pressure.

**Database Sampling:** To prevent overfitting and encourage generalizable improvements, each iteration evaluates agents on only 5 randomly selected databases from the 69-database training set, with 30 randomly sampled questions per database (BIRD supplies a median of 82 questions per database for a total of 6.6K training questions). This deliberate undersampling ensures that agents never see

the complete dataset in any single iteration, forcing the evolution process to discover robust patterns rather than memorizing specific database quirks. The random sampling changes each iteration, exposing agents to different challenges and preventing specialization on a fixed subset.

# 4 EXPERIMENTS

## 4.1 EXPERIMENTAL SETUP

We evaluate on the BIRD benchmark (Li et al., 2023), a large-scale cross-domain dataset with 69 training databases and 11 development databases. Each evolution iteration (using only Haiku 4.5 LLMs for SQL generation) costs approximately $2 in Claude API for the calls and runs for 22 minutes on average, enabling rapid experimentation cycles. Note that evolution is accomplished via Claude Code (using either Sonnet or Opus 4.5 LLMs) and does not incur direct costs, but rather uses up a weekly usage quota. Empirically, each 30-iteration run uses about 12% of the weekly usage quota of a Claude Max plan.[4] Computational requirements are modest. Experiments were conducted on either a MacBook Pro (48GB RAM) or an Azure VM with 8 vCPUs and 32GB RAM.

## 4.2 MAIN RESULTS

We demonstrate RoboPhD's effectiveness by evolving agents on the training set and evaluating them across three tiers of Anthropic models: Haiku-4.5, Sonnet-4.5, and Opus-4.5. As noted above, the `naive` baseline represents the simplest possible agentic design, similar to what someone would use if they were new to the Text-to-SQL problem, with a 50-line Python tool that only dumps raw DDL schema statements (see Appendix E for the complete agent). The naive agent represents the starting point of our evolutionary process as well as our experimental baseline.

Table 1: Results on BIRD dev set. Evolution provides larger improvements on cheaper models.

|  | Opus-4.5 | | Sonnet-4.5 | | Haiku-4.5 | |
|---|---|---|---|---|---|---|
|  | Accuracy | Cost/Query | Accuracy | Cost/Query | Accuracy | Cost/Query |
| Naive | 69.0% | 1.61¢ | 65.7% | 0.56¢ | 57.2% | 0.34¢ |
| Best Evolved | 71.3% | 3.13¢ | 69.2% | 0.87¢ | 66.1% | 0.51¢ |
| Δ | +2.3 | | +3.5 | | +8.9 | |

There are three key takeaways from our main results, shown in Table 1:

(1) Evolution produced the largest gains on the cheapest models. While the evolved Opus agent improves by 2.3 points over naive Opus, the evolved Haiku agent gains 8.9 points—nearly four times the improvement. This inverse relationship between model capability and evolutionary benefit suggests that more powerful models already capture much of what can be learned through prompting and tooling, leaving less room for improvement.

(2) Relative to the naive baseline, evolution delivers higher accuracy at lower cost. An evolved Haiku agent (66.1%, 0.51¢/query) exceeds the accuracy of a naive Sonnet agent (65.7%, 0.56¢/query) at lower cost. Similarly, an evolved Sonnet agent (69.2%, 0.87¢/query) exceeds naive Opus (69.0%, 1.61¢/query) at roughly half the cost. Since evolution is a one-time training cost while inference is ongoing, organizations can deploy cheaper models with evolved prompts rather than expensive models with naive prompts.

(3) Evolution delivers meaningful improvements across all model tiers, arguing for the broad applicability of RoboPhD's approach. Even Opus-4.5, which starts from a strong 69.0% baseline, benefits from evolution, reaching 71.3%—our highest recorded accuracy on the BIRD dev set. As of the time of this writing, 71.3% ranks 17th on the BIRD dev set.[5] We would argue that this is a strong result to achieve without the benefit of human domain knowledge.

---

[4] https://www.claude.com/pricing/max
[5] https://bird-bench.github.io/

## 4.3 EVOLUTION ANALYSIS

**Champion Agent Design.**  Our best-performing agent, `iter18_hybrid_comprehensive_analyzer`, emerged from *cross-pollination*—in the 18th round of evolution, the Evolution AI synthesized techniques from three parent agents that had each reached 76% training accuracy on the 17th iteration through different approaches: adaptive context scaling, semantic pattern analysis, and precision column coverage. Rather than choosing one approach, evolution combined their complementary strengths into a unified tool. The key design innovation was a size-based feature matrix that adjusts analysis depth based on database complexity:

| Feature | Small ($\leq$150 cols) | Medium ($\leq$300 cols) | Large ($\leq$400 cols) | Ultra (>400 cols) |
|---|---|---|---|---|
| Sample values/column | 10 | 5 | 3 | 1 |
| Enum value limit | All | 15 | 5 | 0 |
| Semantic patterns | Full | Essential | Skip | Skip |
| Cross-table validation | Full | Critical | Skip | Skip |

Table 2: Size-adaptive feature matrix. The agent gracefully degrades analysis depth for larger databases to prevent overflowing the 200K token Claude context limit while maintaining comprehensive analysis for typical BIRD databases.

This adaptive approach prevents catastrophic failures on large databases (which previously caused 0% accuracy due to context overflow) while maintaining rich analysis for small and medium databases where detailed semantic understanding improves accuracy.

**Database Analysis Script.** The ∼1000-line Python script generates a structured 10-section analysis: (1) complete schema DDL, (2) table overview with row counts, (3) detailed column analysis with data types and sample values, (4) foreign key relationship map with cardinality, (5) enumerated value reference for low-cardinality columns, (6) value ranges for numeric columns, (7) format detection (dates, currencies, codes), (8) semantic patterns like temporal sequences or hierarchical relationships, (9) cross-table validation identifying orphaned foreign keys, and (10) query guidance with common pitfalls for the specific database.

**Evolved SQL Instructions.** The eval instructions (527 lines) encode patterns discovered across 18 iterations of evolution, including: precise column selection rules (return only columns explicitly requested, not those used for filtering or sorting), evidence interpretation patterns (when evidence says "full address refers to street_num, street_name, city," return all three columns in that exact order), and aggregation guidance (apply LIMIT 1 only for superlatives like "best" or "highest," not for singular nouns).

## 5 DISCUSSION AND FUTURE WORK

### 5.1 PRACTICAL APPLICABILITY

RoboPhD produces easily deployable artifacts. The evolved agent consists entirely of a Python script and a markdown file containing SQL generation instructions—no complex frameworks or dependencies. At deployment time, the Python script runs once per database as an offline preprocessing step, generating a structured analysis without seeing any user questions. This analysis is concatenated with the SQL generation instructions to form a system prompt.

At runtime, answering a user's question requires only this system prompt, the question, and any provided evidence—there is no dependency on Claude Code, the evolutionary infrastructure, or any of the complexity described in this paper. The only additional logic beyond the prompt is the universal verification step (Section 3.2.3), which validates query results and optionally retries. This simplicity enables straightforward integration into production systems.

## 5.2 Architectural Options Under Investigation

RoboPhD's framework supports several capabilities that we have explored but not included in this paper due to insufficient experimental evidence. In the interest of simplicity, we focus on the cross-pollination strategy with tool-only execution, but note these alternatives for future investigation:

**Meta-evolution.** A meta-agent critiques and evolves the evolution strategies themselves, meta-evolving new strategies and scheduling them strategically based on performance trends and review of the reasoning and post-evolution reflection of the evolutionary agents. Our preliminary experiments show promise, but results remain ambiguous—meta-evolution may primarily benefit very lengthy runs exceeding 30 iterations where strategy adaptation becomes more valuable.

**Alternative evolution strategies.** We developed several manually-written strategies beyond cross-pollination, including a *research-driven* strategy that instructs the Evolution AI to study academic papers and extract applicable techniques. While the research-driven strategy is intriguing, we view these alternative strategies as likely superseded by meta-evolution, which could discover the most effective strategies automatically.

**LLM-based database analysis.** Our current approach is "all-or-nothing"—either a deterministic Python script or full LLM reasoning. A promising middle ground would allow generated Python scripts to invoke the LLM in targeted situations where semantic understanding provides high value, such as inferring implicit relationships or disambiguating column purposes.

## 5.3 Broader Applications

**Other Text-to-SQL benchmarks.** While we evaluate on BIRD, the RoboPhD framework is benchmark-agnostic. Spider 2.0 (Lei et al., 2024) presents a natural extension, with its enterprise-scale databases and more complex queries requiring multi-step reasoning. BIRD-Critic (Li et al., 2025b) offers another direction, focusing on debugging and fixing incorrect SQL queries rather than generation from scratch.

**Other domains.** Given that a key feature of the system is an absence of human-injected text-to-SQL domain knowledge, we would expect RoboPhD to generalize beyond Text-to-SQL. Code generation represents a particularly promising application, where similar evolutionary pressure could discover effective prompting strategies and tool designs for programming tasks.

## 6 Conclusion

We presented RoboPhD, a system where AI agents autonomously conduct research to improve Text-to-SQL performance. Our system achieves 71.3% accuracy on the BIRD Dev set through systematic evolution, discovering effective strategies without human intervention in the research loop. Like a tireless PhD student, RoboPhD runs experiments continuously, learning from failures and evolving better approaches.

While our system operates within the bounded domain of database queries, it demonstrates that AI agents can conduct meaningful research cycles—analyzing failures, forming hypotheses, and iteratively refining solutions—without human intervention. The techniques explored here, including competitive evolution and error-driven learning, may extend to other domains where systematic experimentation is feasible. We open-source our framework to enable the community to explore whether similar approaches can accelerate capability development in domains beyond Text-to-SQL.

## Ethics Statement

This work involves autonomous AI systems with significant operational freedom, raising important safety and security considerations. We acknowledge several critical risks:

**Execution Safety:** RoboPhD operates with Claude Code in an automated mode where generated code is executed without human review. While model alignment has reduced risks significantly, autonomous code execution remains inherently dangerous—the system could theoretically execute

destructive commands. We mitigate this through process isolation and filesystem permissions, but acknowledge this remains an open challenge for autonomous AI systems.

**Security Vulnerabilities:** Deploying AI-evolved code introduces novel attack surfaces. The evolved Python tools process database schemas and could potentially be manipulated by adversarial inputs to exfiltrate data or execute unintended operations. While these considerations are not a concern for BIRD's open-source databases, production deployments would require strict sandboxing, network isolation, and comprehensive security auditing of the evolved artifacts.

**Code Release:** We release our code openly to enable reproducibility and community scrutiny. While the techniques demonstrated here are domain-specific, we encourage practitioners deploying similar autonomous code generation systems to implement appropriate safeguards including sandboxing, least-privilege execution, and security review of generated artifacts.

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

## A  LLM USAGE

**Scope and philosophy.**  In keeping with the spirit of this project, we made *extensive* use of large language models (LLMs) throughout. Our goal was to study whether LLMs can function as autonomous research agents with minimal domain-specific guidance from the authors. In addition to using Claude Code to help write, enhance, and refactor the code, we used Claude to help write design documents to brainstorm and clarify new ideas, and Claude helped author parts of the manuscript. In each case there was human oversight and revision applied, especially for the manuscript itself.

## A.1 ROLES WITHIN THE ROBOPHD SYSTEM

**Evolution (Section 3.2.6).** As discussed in Section 3.1, a core contribution is showing that an LLM can serve in the role of a graduate student. Concretely, we used *Claude Code* running *Opus 4.5* or *Sonnet 4.5* for the Evolution Agent. This agent generated new agent packages (prompts and python tools) across iterations.

**SQL Generation (Section 3.2.3).** The SQL Generation phase used *Haiku*, *Sonnet*, or *Opus 4.5* via the Claude API (not via Claude Code) to translate questions (and evidence) into executable SQL, conditioned on the database analysis and evaluation instructions.

**Minimal domain guidance (Sections 3.1, 3.2).** By design, author-provided Text-to-SQL know-how was kept to an absolute minimum. Evolution strategies specify *process* (e.g. cross-pollinate leading agents, analyze errors, etc.) rather than *scientific content*; improvements therefore arise from the models' latent knowledge and error-driven iteration.

## A.2 LLMS USED AS SOFTWARE-ENGINEERING TOOLS

In addition, and consistent with the project's philosophy, we used *Claude Code* (primarily with *Opus* and *Sonnet*) models as a development assistant for the RoboPhD infrastructure. The RoboPhD codebase was almost entirely authored through Claude Code sessions, with the human authors providing specifications, reviewing diffs, enforcing style/tests, and making final design decisions. In this context, Claude Code functioned as a coding tool under close human supervision. Very little of the final codebase was the result of direct manual edits by the authors, with one exception: the evolution strategy prompts discussed in Section 3.2.4 and shown in Appendix C were primarily hand-written. We see this as consistent with the Graduate Student metaphor laid out in Section 3.1: we, the authors, provide thoughtful strategic guidance (similar to the role of a research advisor), while leaving autonomy to the graduate student to iterate towards a solution. As discussed in Section 5, evolving this strategic guidance is a direction for future research.

## A.3 LLMS USED IN MANUSCRIPT PREPARATION

In contrast to the codebase, most of the manuscript text was written by the authors. However, portions of this paper were drafted and edited by the authors with assistance from *Claude Code (Opus 4.1 and 4.5)*, *ChatGPT 5*, and *Gemini 3*. In these cases, the authors determined the structure and arguments. Where LLM-suggested text was used, it was reviewed and, as needed, rewritten by the authors for accuracy and clarity.

## A.4 ATTRIBUTION, AUTHORSHIP, AND ACCOUNTABILITY

Although LLMs played a significant role in ideation (via the Evolution Agent), engineering assistance, and writing support, all scientific claims, experiments, analyses, and conclusions are the responsibility of the human authors. The models did not have independent access to private test data or to submission systems, and they did not make publication decisions.

## A.5 DATA GOVERNANCE AND SAFEGUARDS

LLM prompts and outputs were constrained to benchmark-permitted artifacts (schemas, evidence, and allowed literature). We avoided leaking ground-truth SQL or test questions to the agents; evaluation used standard execution-accuracy protocols. Logs of prompts/outputs for key experiments are retained for audit and reproducibility.

# B REPRODUCIBILITY STATEMENT

The results in this paper are reproducible using the open source code bundle provided as supplementary material to this manuscript. The entire framework can be run from scratch following instructions in the README, which will execute all of the iterations of evolution, reproducing agents similar to the ones featured in the results and discussion earlier in the paper. In the main body of the paper, we note the model families and modes (*Claude Code Opus/Sonnet 4.5* for evolution and *Claude*

*Sonnet 4.5* for evaluation), interface (Claude Code vs. Claude API), and where nondeterminism is relevant, we note seeds and sampling parameters in the released configs. Despite seeds in our code for the randomness that we control, there is some non-determinism that we cannot control in Claude Opus/Sonnet API responses that may lead to minor differences in the results when run from scratch. Given that this is an evolutionary approach over many iterations, small variations in one iteration, may lead to amplified differences in subsequent iterations.

## C  CROSS-POLLINATION STRATEGY PROMPT

This appendix provides excerpts from the cross-pollination evolution strategy that guides the Evolution Agent. The full prompt ($\sim$240 lines) emphasizes process-oriented guidance rather than domain-specific Text-to-SQL techniques.

### C.1  CORE OBJECTIVE

> *"You are creating a new Claude Code agent primarily by combining successful elements from multiple existing agents using the three-artifact architecture."*
> *"\*\*Note:\*\* Although you are primarily using a cross-pollinating approach, you can use a new idea of your own if you think you see an opportunity."*

### C.2  CROSS-POLLINATION APPROACH

The strategy instructs the Evolution Agent to identify complementary strengths:

> *"When examining top-performing agents, ask: Which techniques do different agents use effectively? Which approaches consistently achieve the best results? How do different agents' tools complement each other? What patterns of effectiveness emerge across multiple agents? Which combinations of techniques might create synergies?"*
>
> *"Cross-Pollination Approach: 1) Identify Agent A's most effective techniques. 2) Identify Agent B's complementary strengths. 3) Identify Agent C's unique successful approaches. 4) **Your tool**: Combine these complementary techniques into one comprehensive analyzer."*

### C.3  TOOL-ONLY REQUIREMENT

The strategy mandates deterministic execution:

> *"The system supports a **tool-only execution mode** where your Python/shell tool generates a complete analysis file that is directly copied to the agent output, bypassing the AI agent entirely. This is the REQUIRED approach for this cross-pollination strategy."*

### C.4  META-LEVEL GUIDANCE

The strategy concludes with process-oriented direction:

> *"Your overall goal: Push accuracy higher through tool-only cross-pollination. You are an expert in the field of Text2SQL. Use your knowledge of the field, your analysis of what is bringing accuracy down with current agents, and your analysis of which tool-based techniques from each agent work best to build an agent package that will achieve higher accuracy than previous agents on a set of databases that you haven't seen before."*
>
> *"Remember: **Think harder**[6] than you normally would about this. Review the tools from multiple top-performing agents, identify what makes each one effective, and*

---

[6]"Think harder" is a Claude Code keyword that triggers extended reasoning. See https://www.anthropic.com/engineering/claude-code-best-practices.

> *combine the best elements into a comprehensive tool-only solution* that achieves
> *your goal of pushing accuracy higher."*

These excerpts demonstrate how we provide high-level research methodology—analyze multiple agents, identify complementary strengths, combine deterministic techniques—without injecting domain-specific Text-to-SQL knowledge.

## D  AGENT SELECTION PROTOCOL

This appendix details exceptions to the standard three-agent selection protocol described in Section 3.2.5.

### D.1  STANDARD SELECTION (ITERATIONS 1–11)

Each iteration tests three agents:

1. **Pending winner(s)**: The winner from the previous iteration. If the previous iteration resulted in a tie, all tied agents are pending winners and receive priority slots.

2. **Newly evolved agent**: The agent created by the Evolution AI for this iteration.

3. **ELO-based selection**: Random selection from the top performers by ELO (excluding pending winners and the new agent).

### D.2  LATE-STAGE EXPLORATION (ITERATIONS 12–30)

In the experiments described in this paper, starting at iteration 12, the system probabilistically substitutes non-evolution iterations to increase testing coverage of the accumulated agent population:

- **70% probability**: Standard cross-pollination evolution with 3 agents

- **15% probability**: "Challenger" iteration—no evolution occurs; instead, 4 agents are tested, targeting under-tested agents with above average ELO (ELO $> 1500$, prioritizing non-pending winners with the fewest prior tests)

- **15% probability**: "None" iteration—no evolution occurs; 4 agents are tested via standard ELO-based random selection

The increased agent count (4 vs. 3) in non-evolution iterations compensates for the lack of a newly evolved agent, allowing more thorough evaluation of existing agents. This late-stage exploration helps identify "hidden gems"—agents with strong potential that received insufficient testing in earlier iterations.

## E  THE NAIVE BASELINE AGENT

This appendix presents the complete `naive` baseline agent referenced in Section 4.2. This agent represents the simplest possible starting point—prompts that someone new to the Text-to-SQL problem might write without any domain expertise.

### E.1  DATABASE ANALYSIS AGENT (AGENT.MD)

The database analysis component uses tool-only execution to extract raw schema:

```
---
name: naive
description: Baseline agent that outputs raw DDL schema using tool-only execution
execution_mode: tool_only
tool_command: python tools/extract_schema.py
tool_output_file: tool_output/schema.txt
---
```

```
# Naive DDL Extractor (Tool-Only)

This agent uses deterministic tool-only execution to extract raw database schema.

## Process

1. **Run Schema Extraction Tool**
   ```bash
   python tools/extract_schema.py
   ```

2. **Read and Output Results**
   - Read the generated schema from `tool_output/schema.txt`
   - Write the complete output to `./output/agent_output.txt`

## Error Recovery

If the tool fails:

1. Check `database.sqlite` exists
2. Verify Python environment has sqlite3 library
3. Examine any error messages in `tool_output/`
4. Attempt to run the tool manually to see errors
5. Fall back to manual schema extraction using SQLite CLI:
   ```bash
   sqlite3 database.sqlite ".schema" > output/agent_output.txt
   ```
```

### E.2 SQL GENERATION INSTRUCTIONS (EVAL_INSTRUCTIONS.MD)

The SQL generation instructions are equally minimal:

```
# SQL Generation Instructions

## Core SQL Requirements

Generate clean, executable SQL:
- No markdown formatting or code blocks
- No comments or explanatory text
- Only the SQL statement

## Evidence Handling

Evidence is important!  When evidence is provided, be sure to follow
it very carefully

## SQLite Specifics

- This is a SQLite database.  Be sure to use SQLite syntax

## Remember

Keep it simple. Return exactly what's requested. Follow evidence literally.
```

### E.3 SCHEMA EXTRACTION TOOL (TOOLS/EXTRACT_SCHEMA.PY)

The naive baseline includes a single 50-line Python script that extracts raw DDL:

```python
#!/usr/bin/env python3
"""Baseline schema extractor – extracts raw DDL using sqlite3."""

import sqlite3
import sys

def extract_schema(db_path: str, output_file: str):
    """Extract complete database schema as DDL."""
    try:
        conn = sqlite3.connect(db_path)
        cursor = conn.cursor()

        # Get all schema DDL statements (equivalent to .schema command)
        cursor.execute("""
            SELECT sql || ';'
            FROM sqlite_master
            WHERE sql IS NOT NULL
            ORDER BY tbl_name, type DESC, name
        """)

        schema_statements = cursor.fetchall()

        # Format and write output
        output_lines = []
        for (sql,) in schema_statements:
            output_lines.append(sql)

        schema_output = '\n'.join(output_lines)

        with open(output_file, 'w') as f:
            f.write(schema_output)

        print(f"Schema extraction complete – wrote {len(schema_statements)} DDL state
        conn.close()
        return 0

    except Exception as e:
        print(f"ERROR: {e}", file=sys.stderr)
        return 1

if __name__ == "__main__":
    exit(extract_schema("database.sqlite", "tool_output/schema.txt"))
```