# OpenReview forum: "RoboPhD: Self-Improving Text-to-SQL Through Autonomous Agent Evolution"
_ICLR.cc/2026/Conference — ICLR 2026 Conference Desk Rejected Submission_

### Official Review · Reviewer_NFLz · 2025-10-25

**Soundness:** 2
**Presentation:** 2
**Contribution:** 2
**Rating:** 4
**Confidence:** 4

**Summary:**

The paper proposes RoboPhD, an iterative self-evolving NL2SQL pipeline at test time. It consists of an offline profiling agent to understand databases and schemas, an online generation agent to generate SQL code, and an evolution agent to provide feedback for building new versions of agents. The full system is evaluated with Claude Code and has ~2% absolute improvement compared to baselines.

**Strengths:**

- The method completely relies on test-time optimization, which is a compelling story if one cares about cost.
- The goal of building an autonomous domain independent research agent is ambitious.
- The most part of this paper is well-written and easy to understand.

**Weaknesses:**

- The performance improvement with the proposed method seems to be marginal, especially when compared with other methods in the BIRD-bench leaderboard. Given the scale of the model and complexity of the framework, ~2% absolute improvement doesn’t seem to justify 80 iterations of complex evolution.
- The proposed methodology with prompt optimization and tool evolvement doesn’t seem to be capable of fundamentally resolve the challenges for NL2SQL, e.g. schema linking errors with ambiguous column/table names, logic errors with missing DISTINCT or NOT NULL keywords that are vaguely implied by the users. Those often require deeper understanding on the tables or stronger reasoning capabilities.
- The claim of RoboPhD being an autonomous AI research framework is exaggerated in many ways and the analogy with advisor/graduate student is misleading. I don’t see any examples showing the agent can “conduct systematic research” It is more like a trial-and-error pipeline instead of principled research cycle that can generate innovative ideas. With the proposed algorithm, I am concerned that the LLM may learn superficial experiences from one random batch which might not be generalizable to other batches and test data.
- Random selection on hand-crafted evolution strategies is handwavy and might have limited the performance of the framework.
- Lack of proper quantitative comparison with prompt optimization work since one of the new contributions of this paper lies in co-optimization of prompts and tools. I don’t see how tool evolving improves the performance along with the prompt optimization.

**Questions:**

- It is unclear to me how research-driven evolution strategy improves the agents given many top performers on BIRD-bench leaderboard applies systematic orchestration with different components that might involve multiple fine-tuned embedding and/or generation models (e.g. CHASE-SQL, Arctic-R1 or Reasoning-SQL). Is tool call with python code enough for achieving the goal?
- How to make sure the fix to the errors are generalizable, especially when a lot of NL2SQL errors come from schema linking?
- Can authors provide more details on how refinement strategy works and how it is different from other strategies?

---

> ### Author Response · Authors · 2025-12-04
> **Response to Reviewer NFLz**
>
> Thank you for your thoughtful review. We address each concern below.
>
> ---
>
> ## Weaknesses
>
> ### W1: "~2% absolute improvement doesn't seem to justify 80 iterations"
>
> The revised paper standardizes on 30-iteration runs. More importantly, the key finding is the **inverse relationship between model capability and evolutionary benefit**:
>
> | Model | Naive | Evolved | Improvement |
> |-------|-------|---------|-------------|
> | Opus 4.5 | 69.0% | 71.3% | +2.3 |
> | Sonnet 4.5 | 65.7% | 69.2% | +3.5 |
> | Haiku 4.5 | 57.2% | 66.1% | **+8.9** |
>
> The +8.9 point Haiku improvement is nearly four-fold larger than Opus. This enables "skip a tier" deployment: evolved Haiku (66.1%, 0.51¢/query) exceeds naive Sonnet (65.7%, 0.56¢/query) at lower cost. Each 30-iteration run costs ~$60—a modest investment.
>
> ### W2: "Doesn't seem capable of fundamentally resolving NL2SQL challenges"
>
> We agree fundamental NL2SQL challenges remain open. However, our evolved agents discovered specific techniques:
>
> **Schema linking:** The champion agent uses foreign key analysis and sample values to disambiguate column names, providing concrete examples that enable matching question terms to correct columns.
>
> **NULL handling:** The champion learned when *not* to add IS NOT NULL filters—e.g., "List all reviews" should include NULL values, not filter them out.
>
> **Column ordering:** Evolution discovered that BIRD's evaluation requires precise column ordering—a non-obvious insight that improved accuracy.
>
> These emerged through competitive evolution, not human programming.
>
> ### W3: "The claim of RoboPhD being an autonomous AI research framework is exaggerated"
>
> We have toned down these claims. The revision emphasizes we operate "within the bounded domain of database queries."
>
> However, RoboPhD does conduct systematic research: hypothesis formation (documented in `reasoning.md`), controlled experiments (identical samples), quantitative evaluation (ELO), and iterative refinement at two levels:
>
> * cross-iteration evolution where agents compete across 30 iterations
> * within-session refinement (*Deep Focus*, new to this revision) where the Evolution AI iterates like a graduate student, creating an initial agent, testing it on prior iteration data, analyzing failures, and refining the agent within the same Claude Code session.
>
> Regarding "superficial experiences from one random batch": rotating database samples (5 of 69 per iteration) forces generalization. Overfit agents perform poorly on subsequent iterations. Strong dev set performance validates this.
>
> ### W4: "Random selection on hand-crafted evolution strategies is handwavy"
>
> The revised paper focuses on **cross-pollination with tool-only execution**—a single, well-defined strategy. Weighted random selection is now limited to agent selection (Appendix D), not strategy selection.
>
> ### W5: "Lack of proper quantitative comparison with prompt optimization work"
>
> A full comparison is beyond scope. Key distinctions: (1) RoboPhD co-evolves Python tools (~1000 lines) alongside prompts, generating database-specific context that prompt optimization cannot capture; (2) the evolved tool is fixed at deployment ($0.00/database), unlike methods requiring per-task demonstrations.
>
> ---
>
> ## Questions
>
> ### Q1: "How does research-driven evolution improve agents given top performers use fine-tuned models?"
>
> We now use **cross-pollination**, combining techniques from multiple agents without fine-tuning. Our 71.3% dev accuracy (17th on BIRD) shows pure prompting remains competitive.
>
> ### Q2: "How to ensure fixes are generalizable?"
>
> Three mechanisms: (1) rotating database samples prevent overfitting; (2) question subsampling prevents memorization; (3) held-out dev set (11 unseen databases) validates generalization.
>
> ### Q3: "Can authors provide more details on how refinement strategy works?"
>
> We now focus on cross-pollination. Refinement was a degenerate case (single parent agent), so we simplified to just cross-pollination.
>
> ---
>
> We hope these clarifications address your concerns.

---

### Official Review · Reviewer_Ccc2 · 2025-10-26

**Soundness:** 3
**Presentation:** 2
**Contribution:** 2
**Rating:** 4
**Confidence:** 3

**Summary:**

This paper introduces RoboPhD, a novel framework where AI agents autonomously conduct research to improve their own performance on the Text-to-SQL task. The system employs a closed-loop evolutionary cycle composed of three coordinated agents: a Database Analysis agent, a SQL Evaluation agent, and an Evolution agent. The Evolution agent uses a portfolio of strategies (e.g., refinement, error-analysis, and adapting insights from academic papers) to generate new, improved agents. A key component of the framework is an ELO-based selection mechanism, which manages a competitive ecosystem of agents, enabling "survival-of-the-fittest" dynamics. Starting from a simple baseline, the system autonomously evolves agents that discover sophisticated Text-to-SQL strategies, achieving a significant, measurable performance gain and reaching a strong 68.6% execution accuracy on the BIRD test benchmark.

**Strengths:**

1. The primary strength of this paper is the innovative concept of an "AI researcher" framework that automates the cycle of hypothesis, experimentation, and refinement. The three-agent architecture and the use of an ELO rating system for agent selection are elegant and technically sound.
2. The paper demonstrates that its framework can autonomously achieve a measurable performance gain (~2-2.6% absolute accuracy) over baselines on a challenging benchmark. This provides concrete evidence for its central claim.
3. The case study on the champion agent's evolution (Section 4.3) is also a strength. It provides a clear narrative of how the system learns, adding a deep layer of credibility and insight that goes beyond just reporting final scores.

**Weaknesses:**

1.  The most significant weakness is the lack of systematic ablation studies on the evolution strategies. The authors acknowledge this (lines 450-452), but without this analysis, it is difficult for the reader to understand the marginal contribution of each component (e.g., "Research-Driven" vs. "Error-Focused"). This is a critical piece of analysis needed to fully validate the framework's design.

2. The framework's success is demonstrated on a single, high-end proprietary model (Claude Opus 4.1). This raises questions about whether the autonomous research capability is a generalizable property of the framework or highly dependent on the specific, latent abilities of this particular LLM.

3. The inherent stochasticity of the evolutionary process and LLM API calls, while acknowledged by the authors, may pose challenges for exact replication of the results and the specific evolutionary paths discovered.

**Questions:**

1. To strengthen the paper, could you provide an ablation study on the evolution strategies? For instance, what is the final performance if the "Research-Driven" strategy is disabled? This would help isolate the contribution of de-novo discovery versus literature adaptation and is crucial for increasing my confidence in the soundness of the specific framework design.

2. How sensitive is the RoboPhD framework to the choice of the underlying LLM for the Evolution Agent? Have you conducted any preliminary experiments with other models (e.g., leading open-source models like Llama-3-70B) to assess if this autonomous research capability is a general property of current SOTA models? Answering this would greatly clarify the scope and generalizability of your contribution.

3. The use of the ELO system is well-justified for handling non-transitivity. Can you provide a concrete example of a "rock-paper-scissors" dynamic observed during your experiments? This would powerfully illustrate the practical necessity of this component.

---

> ### Author Response · Authors · 2025-12-04
> **Response to Reviewer Ccc2**
>
> # Response to Reviewer Ccc2
>
> Thank you for recognizing the innovative concept of an "AI researcher" framework and the elegance of the ELO-based selection mechanism. We address each concern below.
>
> ---
>
> ## Weaknesses
>
> ### W1: "Lack of systematic ablation studies on the evolution strategies"
>
> The revised paper addresses this by simplifying the framework. We now focus on a **single evolution strategy: cross-pollination with tool-only execution**. The research-driven and error-focused strategies mentioned in the original submission have been removed from the main results. This simplification provides clarity: the gains come from cross-pollination of successful techniques across agents, not from a complex portfolio of strategies.
>
> ### W2: "Framework's success demonstrated on a single, high-end proprietary model"
>
> The revised paper now includes results across **three model tiers**:
>
> | Model | Naive | Evolved | Improvement |
> |-------|-------|---------|-------------|
> | Opus 4.5 | 69.0% | 71.3% | +2.3 |
> | Sonnet 4.5 | 65.7% | 69.2% | +3.5 |
> | Haiku 4.5 | 57.2% | 66.1% | **+8.9** |
>
> A key finding is the **inverse relationship between model capability and evolutionary benefit**: Haiku gains nearly four times more than Opus. This suggests the framework is not dependent on a single high-end model—in fact, it provides the largest value for cheaper models.
>
> We have not yet tested with open-source models like Llama-3, which we acknowledge as a limitation and direction for future work.
>
> ### W3: "Stochasticity poses challenges for exact replication"
>
> We acknowledge this limitation. We provide seeds for randomness we control (database selection, question sampling), but LLM API non-determinism is inherent. The evolutionary nature means small variations can compound across iterations.  It is encouraging that we found models that yielded gains over the naive baseline with three Claude models for SQL generation.  This shows a degree of consistency in the results.
>
> ---
>
> ## Questions
>
> ### Q1: "Ablation study on evolution strategies?"
>
> As noted in W1, the revised paper simplifies to a single strategy (cross-pollination). We found that refinement was effectively a degenerate case of cross-pollination (single parent), and research-driven adaptation added complexity without clear benefit. The streamlined design makes ablation less necessary—the contribution is the evolutionary framework itself, not a specific strategy portfolio.
>
> ### Q2: "Sensitivity to underlying LLM for Evolution Agent?"
>
> The impact of the strength of the model for SQL generation is quite significant as shown by the results in Table 1 of the revised paper.  For instance, we see an 8.5\% gain in accuracy for the Sonnet naive model over Haiku and a 3.3\% gain for Opus over Sonnet.
>
> However the impact of model strength on evolution is not as large.  Our strongest overall model, achieving 71.3\% accuracy with Opus 4.5 inference on the BIRD development set, used Sonnet 4.5 as the Evolution model.  Our strongest models for Sonnet and Haiku used Opus 4.5 as the Evolution model (as one would expect), but interestingly, a run with Haiku as the Evolution model and Opus as the experimental meta-evolution model delivered the third highest accuracy for both the Opus and Haiku inference use-cases.
>
> We have not tested with open-source models. Given that cross-pollination requires analyzing multiple agent codebases (~1000 lines each) and synthesizing improvements, we expect this capability requires frontier-class models, but this remains to be validated.  We would note, though, that the cost of evolution is a one time offline cost.  In an industrial setting, inference costs dominate.
>
> ### Q3: "Concrete example of rock-paper-scissors dynamic?"
>
> See Section 3.2.5 of the revised paper where we clarify this dynamic.
>
> ---
>
> We hope these clarifications address your concerns. The key revision is simplification to a single evolution strategy with multi-model validation.

---

### Official Review · Reviewer_kENo · 2025-10-27

**Soundness:** 1
**Presentation:** 1
**Contribution:** 2
**Rating:** 0
**Confidence:** 5

**Summary:**

The paper introduces RoboPhD, an autonomous AI agent system designed to improve Text-to-SQL performance without human intervention. The system integrates three agents, including a Database Analysis Agent that profiles database schemas offline, a SQL Evaluation Agent that generates executable SQL queries, and an Evolution Agent that designs new agents based on performance feedback. The Evolution Agent employs ELO-based evolutionary selection and multi-strategy weighted random sampling to guide exploration across refinement, research-driven, and error-focused strategies. The system achieves 68.6% execution accuracy on the BIRD benchmark.

**Strengths:**

1. The closed-loop system combines evolution, evaluation, and analysis agents, allowing agents to learn from their own experiments without human intervention.

2. The use of ELO ratings as an active selection mechanism in evolutionary optimization is well justified.

3. The core algorithm (Algorithm 1: RoboPhD Evolution Cycle) is clearly written and well described.

**Weaknesses:**

1. Experiments are only conducted on very simple baselines, text-to-SQL literature is not considered. The current SOTA text-to-SQL method has achieved more > 80% execution accuracy on the BIRD leaderboard [1]. I don't want to be harsh, but this work is obviously far away from latest text-to-SQL research, when the cost is also high.

2. In addition, the evaluation is very shallow.  Experiments are conducted only on a single benchmark, limiting the generalizability of the claims. The evolution analysis in sec 4.3 only provides a case study. The paper mentioned that the detailed ablation study will be provided in "forthcoming extended version", which is also very weird.

3. There are many overclaimed "big words" in the paper writing (e.g., bullets in the introduction). The main contribution of this paper is not clear.

[1] https://bird-bench.github.io/

**Questions:**

I highly recommend that this paper include related text-to-SQL works in the evaluation. And highlight the core novelty and contribution under the literature.

---

### Official Review · Reviewer_vC5u · 2025-11-03

**Soundness:** 2
**Presentation:** 2
**Contribution:** 1
**Rating:** 2
**Confidence:** 4

**Summary:**

The paper presents RoboPhD, a framework designed to autonomously improve Text-to-SQL performance through iterative agent evolution. The system includes a database analysis agent, an SQL generation agent, and an evolution agent that refines the others based on ELO-ranked performance feedback. The process aims to mimic a “PhD-style” research loop where agents propose, test, and refine hypotheses. Experiments on the BIRD benchmark show an improvement from around 66% to 68.6% execution accuracy after 80 iterations. The authors claim that this demonstrates a step toward autonomous research and self-improving AI systems.

**Strengths:**

- Creative and well-written. The “autonomous PhD” metaphor and clear structure make the paper engaging and easy to follow.
- Complete system implementation. The framework runs end-to-end with concrete results and released configurations, showing technical feasibility.
- Innovative use of ELO-based evolution. Adapting ELO scoring for agent selection is a neat and transferable idea.
- Potentially inspiring direction. The paper contributes to the growing discussion on self-improving LLM systems and agentic evolution.

**Weaknesses:**

- Insufficient experiments and analysis. Evaluation is limited to a single dataset (BIRD) and two short runs, with no ablations, variance reporting, or comparisons to prompt-optimization baselines (e.g., DSPy, OPRO). The 2% improvement could easily stem from randomness.
- Overstated claims. The work demonstrates automated prompt search within a fixed pipeline, not genuine autonomous “research.” The conceptual framing exceeds what the evidence supports.
- Limited novelty beyond Text-to-SQL. Similar self-evolving or self-optimizing agent frameworks have been proposed in other domains (e.g., AI Scientist, TextGrad, LLM-as-Optimizer). The novelty lies mainly in applying these ideas to Text-to-SQL.
- Lack of analytical depth. The evolution analysis is narrative and anecdotal (iter2 -> iter12 -> iter60) without quantitative explanations or systematic error breakdowns.
- No cross-domain validation. The framework’s generality remains untested; it is unclear whether the method extends beyond the BIRD setting.

**Questions:**

The experimental evidence is too limited to support its strong claims of autonomy and self-improvement. I'd like to see the experimental results on newer datasets like Spider 2.0 or BIRD Interact

---

### Note · Program_Chairs · 2026-01-17
**Submission Desk Rejected by Program Chairs**

The following references in this submission do not refer to real documents and/or have major errors in bibliographic information:



1. Ahmet Dönder et al. Genasql: A generative agent for advanced sql generation. arXiv preprint arXiv:2505.14174, 2025
2. Mohammadreza Pourreza et al. Reasoning-sql: Distilling reasoning capabilities from llms to smaller models for text-to-sql. arXiv preprint arXiv:2503.23157, 2025.
3. Yunnan Sheng et al. Csc-sql: Community-driven schema construction for text-to-sql. arXiv preprint arXiv:2505.13271, 2025
4. Jacob Steinhardt. Ai capabilities and alignment research. AI Alignment Forum, 2022.
5. Daniel Maamari et al. Distillery: A compositional framework for text-to-sql. arXiv preprint arXiv:2408.07702, 2024.
6. Hao Li et al. Omnisql: Database systems enhancement with large language models. arXiv preprint arXiv:2503.02240, 2025a.
7. Jianmin Xie et al. Opensearch-sql: Context-aware sql generation with an open-source suite. arXiv preprint arXiv:2502.14913, 2025.
8. Vladislav Shkapenyuk et al. Askdata: Naturally leveraging data analysis agents for complex text-to-sql generation. arXiv preprint arXiv:2505.19988, 2025.